# Fatty Acid Excess Dysregulates CARF to Initiate the Development of Hepatic Steatosis

**DOI:** 10.3390/cells12071069

**Published:** 2023-04-01

**Authors:** Kamrul M. Hasan, Meher Parveen, Alondra Pena, Francisco Bautista, Juan Carlos Rivera, Roxana Ramirez Huerta, Erica Martinez, Jorge Espinoza-Derout, Amiya P. Sinha-Hikim, Theodore C. Friedman

**Affiliations:** 1Division of Endocrinology, Metabolism and Molecular Medicine, Department of Internal Medicine, Charles R. Drew University, Los Angeles, CA 90059, USA; 2David Geffen School of Medicine, University of California, Los Angeles, CA 90095, USA; 3California State University Dominguez Hills, Carson, CA 90747, USA

**Keywords:** CARF (CDKN2AIP), lipo-toxicity, hepatic steatosis, ER-stress, oxidative stress, metabolic stress, NAFLD

## Abstract

CARF (CDKN2AIP) regulates cellular fate in response to various stresses. However, its role in metabolic stress is unknown. We found that fatty livers from mice exhibit low CARF expression. Similarly, overloaded palmitate inhibited CARF expression in HepG2 cells, suggesting that excess fat-induced stress downregulates hepatic CARF. In agreement with this, silencing and overexpressing CARF resulted in higher and lower fat accumulation in HepG2 cells, respectively. Furthermore, CARF overexpression lowered the ectopic palmitate accumulation in HepG2 cells. We were interested in understanding the role of hepatic CARF and underlying mechanisms in the development of NAFLD. Mechanistically, transcriptome analysis revealed that endoplasmic reticulum (ER) stress and oxidative stress pathway genes significantly altered in the absence of CARF. IRE1α, GRP78, and CHOP, markers of ER stress, were increased, and the treatment with TUDCA, an ER stress inhibitor, attenuated fat accumulation in CARF-deficient cells. Moreover, silencing CARF caused a reduction of GPX3 and TRXND3, leading to oxidative stress and apoptotic cell death. Intriguingly, CARF overexpression in HFD-fed mice significantly decreased hepatic steatosis. Furthermore, overexpression of CARF ameliorated the aberrant ER function and oxidative stress caused by fat accumulation. Our results further demonstrated that overexpression of CARF alleviates HFD-induced insulin resistance assessed with ITT and GTT assay. Altogether, we conclude that excess fat-induced reduction of CARF dysregulates ER functions and lipid metabolism leading to hepatic steatosis.

## 1. Introduction

Hepatocytes are crucial for metabolic and non-metabolic functions, including hepatic triglyceride (TG) metabolism. Hepatocytes process free fatty acids (FFAs) and lipids to perform these physiologic functions, with leftover lipid stored as TG in the cytoplasm as droplets [1]. TG storage in the hepatocytes depends on the balance between the rate of acquisition of FFA uptake from the circulation and de novo lipogenesis (DNL) [2] and the rate of breakdown by β-oxidation and secretion as very low-density lipoprotein (VLDL). In some physiologic abnormalities, such as excessive caloric intake, the DNL and uptake of FFAs exceed the utilization by breakdown and secretion, leading to an excess accumulation of TG and initiation of the development of hepatic steatosis, similar to the human condition of non-alcoholic fatty liver disease (NAFLD) [3]. In humans, the prevalence of NAFLD is increasing with the obesity epidemic, and approximately 30% of the American population has NAFLD. Almost 57% of obese and 70% of T2DM patients have NAFLD, which increases to 90% in patients with morbid obesity [4,5,6,7]. Given that current therapies to treat NAFLD and the more severe condition, non-alcoholic steatohepatitis (NASH), are limited, detailed knowledge about the hepatic lipid metabolism in altered physiological settings such as obesity and insulin resistance (IR) is essential to developing an effective therapy.

CARF, also known as CDKN2AIP, has been cloned as ARF (alternate reading frame) binding protein and reported to activate p53 pathways through ARF-dependent and independent manners [8,9,10,11]. Recent studies implicated CARF as a positive regulator of the Wnt-β-catenin signaling pathway that has been shown to contribute to epithelial-mesenchymal transition in cancer [12,13], organ development in zebra fish [14], spermatogenesis in mice [15], and cancer stem cell renewal [16]. Interestingly, CARF expression is low in p53-wt cells compared to p53-mutated or null cells, suggesting that CARF and p53 exhibit an inverse relationship [10,11]. This reciprocal relationship determines the cellular fate in response to various stresses, including oxidative, replicative, oncogenic, DNA damage, and others, leading to either growth arrest, senescence, or apoptosis [17,18]. However, the role of CARF in caloric excess leading to metabolic stress in hepatocytes remains unknown. In this study, we aimed to explore the impact of metabolic stress on hepatic CARF expression and its physiological significance in terms of metabolic diseases, such as NAFLD. 

By using in vitro and in vivo models of hepatic steatosis, global transcriptome analysis, and AAV-mediated overexpression of CARF, we unveil a novel role of CARF in the process of hepatic steatosis. 

## 2. Materials and Methods 

### 2.1. Mice

All animal studies were conducted according to a protocol approved by the Institutional Animal Care and Use Committee of Charles Drew University (IACUC #CDU 22458-03). Six-week-old C57BL/6 male mice were used in this study and were purchased from Jackson Laboratories (Bar Harbor, ME, USA). Mice were kept on a 12-h-light-dark cycle starting from 0600 to 1800 and fed on a normal chow diet (NCD) (D12450B, 10% Fat, 70% carbohydrates, and 20% protein, Research Diets Inc. New Brunswick, NJ, USA) and water ad libitum. To establish a diet-induced obesity (DIO) and hepatic steatosis model, mice (*n* = 10) were fed a high-fat diet (HFD) (Research Diet, D12492, 60% Fat, 20% carbohydrate, and 20% protein) beginning at the age of 7 weeks for 16 weeks. For rescue experiments, the Adeno-associated virus (AAV-9)-mediated gene expression method was used to overexpress CARF in the mouse livers. Intraperitoneal (IP) delivery of AAV-9 was found to be highly efficient in hepatic gene expression in mice [19]. Mice fed on HFD were randomly divided into two groups (*n* = 10). One group of mice was injected IP with 100 µL of PBS containing 10^12^ particles with AAV-empty vector and the other group with AAV-CARF overexpressing vector (Genecopoeia, Rockville, MD, USA) at day 0 of HFD feeding following the protocol described above [19]. Body weight was recorded weekly. After 14 weeks, mice (*n* = 6) were fasted for 8 h and fasting glucose was measured. After that, mice were injected IP with glucose (2 g/kg BW) for a glucose tolerance test (ipGTT). Blood glucose level was monitored at 0, 15, 30, 60 and 120 min post-injection of glucose using Accu-Chek guide glucose strips (Roche, Corona, CA, USA). For the insulin tolerance test (ipITT), after 6 h fasting, fasting blood glucose was measured. Then, insulin at 0.75 U/kg body weight was injected IP, and glucose content was assayed at 0,15, 30, 60, and 120 min with GTT assay. After 16 weeks, mice were fasted for 8 h and sacrificed by decapitation. Livers and fasted serum were collected for further use.

### 2.2. Cell Culture and Treatment 

HepG2, human hepatoma cells (ATCC, Manassas, VA, USA), were cultured in DMEM (ATCC) supplemented with 10% FBS (Gibco, Grand Island, NY, USA) and 1% penicillin and streptomycin (Invitrogen, Carlsbad, CA, USA). Although HepG2 cells have been reported to show some chromosomal [20] and oncogenic gene abnormalities [21,22], they show many of the metabolic characteristics of primary hepatocytes [23,24,25]. To mimic steatosis, in vitro HepG2 cells were treated with different concentrations of palmitate-BSA conjugate (Sigma Aldrich, St. Louis, MO, USA) [26]. Fat-free bovine serum albumin (BSA) (Sigma Aldrich, St. Louis, MO, USA) was used as vehicle control. In addition, normal mouse hepatocyte AML12 (ATCC) was used. AML12 was grown and maintained in DMEM -F12 supplemented with 10% FBS, ITS (10 µg/mL Insulin, 5.5 µg/mL -Transferrin and 5ng/mL selenium) (Gibco, USA), and 40 ng/mL dexamethasone (Sigma Aldrich). For experiments, AML12 cells were grown in DMEM-F12 medium supplemented with 10% FBS.

### 2.3. H and E Staining and Histological Analysis of Liver Samples

Portions of the livers collected from the mice were fixed with 10% formalin and embedded in paraffin. Five μm sections were cut and stained with hematoxylin and eosin (H&E). The stained sections were dehydrated in ethanol series, sealed with xylene, and mounted with coverslip by adding mounting solution (Vector Laboratories, Newark, CA, USA). Sections were photographed using a light microscope (Olympus, DP71, Tokyo, Japan) under a bright field. Quantitative analysis of hepatic steatosis was scored in a blind fashion using a scoring method as described previously [27]. The score was calculated based on the prevalence of macrovesicular steatosis and microvesicular steatosis [27]. The histological features were analyzed with light microscopy Olympus Bx51 (Olympus, Tokyo, Japan) at 10× or 40× objectives in five different fields.

### 2.4. Modulation of CARF Expression in HepG2 Cells

To downregulate the expression of CARF in HepG2 cells, lentiviral Sh-CARF and control Sh-scramble (Sh-SCR) vectors were purchased from Sigma Aldrich (mission ShRNA, Sigma Aldrich, St. Louis, MO, USA). For overexpression of human CARF, a lentiviral-based expression vector was purchased from Genecopoeia, Rockville, MD, USA). Lentivirus packaging was performed in the UCLA vector core following a standard protocol. To downregulate CARF, cells were infected with Sh-CARF and Sh-SCR particles overnight, the medium changed the next morning, and cells were harvested 72 h after infection for further processing. Stable CARF overexpressing (CARF-OE) HepG2 cells were established by infecting HepG2 cells with CARF expressing viral particles and treated with puromycin (2.5 µg/mL) for about 10 days. Expression of CARF was confirmed via immune-blotting assay. 

### 2.5. Oil Red O Staining

HepG2 cells were fixed in 10% neutral formalin in PBS for 24 h and stored in 70% ethanol at 4 °C. After fixation and washing with PBS, cells were rinsed with 60% ethanol and dried completely. Then, 1 mL of Oil-Red-O dye was added per well, incubated for 10 min, and washed with water to remove residual stains. Images were captured using a light microscope (Olympus, DP71, Tokyo, Japan). 

### 2.6. Triglyceride Assay 

Cellular or hepatic triglyceride levels were measured following a protocol provided by a Triglyceride assay kit (Sigma Aldrich, St. Louis, MO, USA). In brief, 10^6^ cells were homogenized in 5% NP40 buffer and subjected to repeated steps of heating at 80–100 °C for 3–5 min and cooled to room temperature. After brief centrifugation, the supernatant was collected, and an aliquot was used to measure TG levels. Hepatic TG was measured using 25–30 mg of liver tissues collected from control and CARF-overexpressing mice (*n* = 6) following the instructions provided in the kit (Sigma Aldrich, St. Louis, MO, USA).

### 2.7. Quantitative Real-Time PCR

Using an RNAase mini kit (Qiagen), total RNA was extracted from HepG2 cells 72 h after knockdown of CARF by Sh-CARF. Sh-scramble (Sh-SCR)-treated cells were used as control. The RNA concentration was determined with a NanoDrop spectrophotometer (Thermo Scientific, Waltham, MA, USA), and quality was assessed with RNA integrity assay (RIN) using Bioanalyzer 2100 (Agilent Technologies, Santa Clara, CA, USA). cDNA was synthesized using Superscript III reverse transcriptase (Invitrogen, 18080–051). Quantitative real-time qPCR was performed with SYBR Green probes using the Applied Biosystems 7900HT Fast Real-Time PCR system. Results were expressed as the fold change in transcript levels. List of primers used in this study was shown in Appendix A.

### 2.8. RNA Sequence Analysis

The total RNA extracted from HepG2 cells as described above were sent to UCLA Technology Center for Genomics and Bioinformatics core for RNA-seq analysis. The quality of the RNA was assayed via RIN using the Bioanalyzer 2100 (Agilent Technologies, Santa Clara, CA, USA). Libraries for RNA-seq were prepared with the KAPA Stranded RNA-Seq Kit. Differentially expressed genes were identified using the edgeR program. Differentially expressed genes were determined according to those genes showing altered expression with a *p* value < 0.05 and more than 1.5-fold changes compared to scramble control. The pathway and network analyses were performed using Ingenuity Pathway Analysis (IPA) software. IPA computes a score for each network according to the fit of the set of supplied focus genes. These scores indicate the likelihood of focused genes belonging to a network versus those obtained by chance. A score of greater than 2 indicates a less than 99% confidence, suggesting that a focus gene network was not generated by chance alone. The canonical pathways generated via IPA are the most significant for the uploaded data set. Fischer’s exact test with the false discovery rate option was used to calculate the significance of the canonical pathway. 

### 2.9. Oxidative Stress or ROS Assay

ROS in cells were determined via staining with a fluorogenic dye H2DCFDA following the protocol provided with the kit (Abcam, Waltham, MA, USA). A total of 15,000 cells were seeded in 96-well plates overnight prior to the assay. After washing with 1× wash buffer, 150 µL of staining solution (20 µM of H2DCFDA in wash buffer) was added to each well and incubated at 37 °C for 45 min. After washing twice with washing buffer, images were captured using a fluorescent microscope (Carl Zeiss, Germany). Fluorescent intensity was measured using Image Pro software V.10. 

### 2.10. Protein Analysis

Western blotting was performed with hepatic tissues (*n* = 6) and HepG2 cells using T-per and M-per lysis buffer (Thermo Scientific, Carlsbad, USA), respectively, as described previously [28,29]. Briefly, 50 mg of mouse liver tissue was lysed via homogenization in 0.5 mL of T-per lysis buffer. HepG2 cell pellets were lysed in 50–100 µL of M-per lysis buffer. Lysates were electrophoresed on a 4–15% gradient gel and transferred onto PVDF membranes using the wet transfer method. After incubation with primary antibodies overnight, blots were probed with horse radish peroxidase (HRP)-conjugated secondary antibodies for 2 h. Results were detected by scanning the blots with the Licor-Odyssey XF imaging system (Licor Biotechnology, Lincoln, NE, USA) after incubation with the Super signal West-Pico chemiluminescent (ECL) substrate (Pierce, Thermo Scientific). Antibodies used in this study were as follows: rabbit polyclonal CARF (A303-861A, Bethyl Laboratories, at 1:2000 dilution), mouse monoclonal AMPK (# 66536-1, Proteintech, at 1:2000 dilution), rabbit polyclonal pAMPK (ab133448, Abcam, at 1:2000 dilution), rabbit polyclonal ACC (# 3676, Cell Signaling, at 1:1000 dilution), rabbit polyclonal pACC (# 3661, Cell Signaling, at 1:1000 dilution), rabbit polyclonal IRE1α (# 3294, Cell Signaling, at 1:1000 dilution), rabbit polyclonal BiP or Grp78 (# 3177, Cell Signaling, mouse polyclonal CHOP (# 2896, Cell Signaling, at 1:1000 dilution), rabbit polyclonal Sirt1 (ab189494, Abcam, at 1:2000 dilution), rabbit polyclonal HO-1 (ab52947, Abcam, at 1:2000 dilution), rabbit polyclonal GPX3 (ab256470, Abcam, at 1:2000 dilution), and mouse monoclonal GAPDH (MAB 374, Millipore-Sigma, at 1:5000 dilution). 

### 2.11. Preparation of Palmitate-BSA Conjugate

BSA-conjugated palmitate was prepared via modification of the seahorse protocol (Seahorse Bioscience, Santa Clara, CA, USA). A 5 mmol/L palmitate (Sigma-Aldrich, St. Louis, MO, USA) solution was prepared at 70 °C. After dissolving, the solution was complexed with 10% fatty-acid-free BSA (Sigma-Aldrich, St. Louis, MO, USA) at an 8:1 palmitate-to-BSA-molar ratio. Then, the BSA-conjugated palmitate solution was added to cell culture medium to achieve a various palmitate concentration as indicated. 

### 2.12. TUNEL Staining

Apoptosis was detected using the DeadEnd Fluorometric TUNEL system (Promega, Madison, WI, USA) following the manufacturer’s instructions. Briefly, HepG2 cells were grown on coverslips in 12-well plates. After treatment, cells were fixed with 10% formalin and permeabilized with 0.1% Triton X-100 added in PBS for 10 min. After permeabilization, cells were washed twice with PBS. Then, cells were incubated with labeling solution with terminal deoxynucleotidyl transferase (TdT) enzyme for 1 h at 37 °C. The cells were then washed with PBS. Finally, anti-fade mounting medium with DAPI (Vector labs) was added and apoptotic cells were observed and counted with a microscope (Carl Zeiss, Jena, Germany).

### 2.13. Statistical Analysis

Statistical analyses were performed using GraphPad Prism 9.4 (GraphPad Software, Inc., San Diego, CA, USA). Results are represented as mean ± SD. qPCR data. Immunohistochemistry quantifications and biochemical data from the experimental groups were evaluated for significance using the parametric and unpaired two-tailed Student’s *t*-tests. One way ANOVA followed by the Holm–Sidak method were used for multiple comparison. All in vitro experiments were conducted at least 3 times, and in vivo experiments were conducted with 10 mice per group. A *p* value < 0.05 was considered significant for all analyses. Significant differences between experimental groups were * *p* < 0.05, ** *p* < 0.01, or *** *p* < 0.001.

## 3. Results

### 3.1. Expression of CARF Was Reduced in Fatty Livers and Cellular Model of Steatosis

To explore the role of CARF in NAFLD, we examined the level of CARF expression in fatty liver samples collected from DIO mice. Compared to NCD controls (Figure 1A), H and E staining from HFD-fed mice showed a marked increase in hepatic fat accumulation (Figure 1B) featuring both microvesicular and macrovesicular steatosis. Quantitative analysis showed a higher NAFLD score in HFD-fed mice in comparison to NCD-fed mice (Figure 1C). Compared to NCD-fed mice, the expressions of pAMPK, pACC, and Sirt1 decreased in HFD-fed livers (Figure 1D,E). Intriguingly, the expression of CARF significantly decreased in HFD-fed livers relative to NCD-fed control mice (Figure 1D,E). To further substantiate these findings, we established an in vitro model of steatosis by challenging HepG2 (a hepatoma cell line) and AML12 (a normal mouse hepatocyte cell line) cells with palmitate, which was reported to show a higher association with de novo lipogenesis over oleic or palmitoleic acid in type 2 diabetes [30]. We treated HepG2 and AML12 cells with 750 µM of palmitate, which is identical to the fasting FFA level in serum of the NAFLD patients [31,32]. In agreement with our in vivo observation, we found that palmitate exposure triggered the reduction of CARF along with Sirt1, pACC, and pAMPK in both HepG2 (Figure 1F,G) and AML12 cells (Figure 1H,I). These results suggest that ectopic fat accumulation reduced CARF expression both in the liver as well as in HepG2 and AML12 cells.

### 3.2. Ablation of CARF Exacerbates Fat Accumulation in HepG2 and AML12 Cells

Downregulation of CARF in both in vivo and in vitro models of hepatic steatosis raises the possibility that loss of function of CARF might regulate hepatic fat accumulation. To address this hypothesis, we knocked down CARF in HepG2 and AML12 cells by infecting lentiviral particles bearing Sh-CARF RNA and Sh-SCR RNA and observed that expression of CARF was substantially reduced by CARF-specific ShRNA (Figure 2A,F, respectively). To demonstrate the effect of CARF knockdown on endogenous levels of lipid, both HepG2 and AML12 cells were stained with fluorescent tagged neutral lipid (Bodipy 494/503). In Figure 2B,G, microscopic images showed the distribution of lipid droplets around the nucleus of the HepG2 and AML12 cells. As expected, fluorescent intensities of the lipid droplets in CARF-depleted cells were much higher with larger lipid droplets than in the scramble-RNA control cells, suggesting that lipid contents increased in the absence of CARF in hepatocytes. In agreement with this, TG levels significantly increased in CARF-depleted cells (Figure 2C,H). These results indicate that fat accumulation was induced by the downregulation of CARF.

Given that CARF expression goes down in both in vivo as well as in vitro models of hepatic steatosis, we assumed that CARF has a protective role against fat deposition in hepatocytes. To address this question, we investigated the effect of gain of function of CARF on fat accumulation in HepG2 cells. We overexpressed CARF using a lentiviral-based gene expression vector and measured TG in HepG2 cells. As shown in Figure 2D, compared to vector control, CARF was expressed approximately two-fold higher in CARF-expressing LV-transduced cells. TG assay revealed that compared to the vector control, overexpression of CARF reduced the endogenous TG levels in HepG2 cells (Figure 2E). Together, these results suggest that CARF has a lipid-lowering effect in hepatocytes. 

### 3.3. CARF Protects against the Ectopic Fat Accumulation

In addition to DNL, FFA released from adipose tissue lipolysis significantly contributes to the development of hepatic steatosis [33]. Therefore, we explored whether CARF has the ability to reduce ectopic fat accumulation in hepatic cells. To address this possibility, we overexpressed CARF in HepG2 cells by transducing lentiviral particles with CARF expression vector. Cells transduced with empty vector were used as control. Vector-control and CARF-overexpressing cells were treated with palmitate-BSA conjugate or BSA only as a vehicle control for 24 h. Oil -Red-O staining showed large lipid droplets with increased staining in palmitate-treated vector control cells, but not in BSA-only treated controls cells (Figure 3A left panel). Consistently, TG levels significantly increased in palmitate-treated vector control cells (Figure 3B). In contrast, CARF overexpression attenuated the accumulation of lipid droplets as well as TG levels in HepG2 cells (Figure 3A,B). Overall, we conclude that CARF lowers ectopic fat accumulation in hepatocytes. 

### 3.4. Deficiency of CARF Induces ER Stress in HepG2 and AML12 Cells

To delineate the underlying mechanisms of the lipid-lowering effect of CARF, we performed global gene expression analysis of HepG2 cells with and without ShRNA-mediated silencing of CARF using RNA sequence analysis. CARF target genes were identified with a set false discovery rate (FDR; <0.05 and fold change ≥ 2). We found that 1057 genes were differentially expressed (DEG) in CARF-deficient cells (Figure 4A). Of these, 225 genes were significantly upregulated, and 835 genes were downregulated (Figure 4A). To identify which pathways were altered in CARF-depleted cells, we performed KEGG and Reactome enrichment analysis for DEGs. We found that many different pathways were affected in the absence of CARF (Figure 4B; Appendix A). Notably, the majority of the DEGs in the “protein processing in ER” or unfolded protein response (UPR) were enriched as the targets of CARF in HepG2 cells. Furthermore, GSEA analysis of DEGs identified that genes associated with classical ER stress signaling, such as GRP78, IRE1α, CHOP, NuPR1, and others, were upregulated in CARF-depleted cells (Figure 5A,B and Appendix A). RT-qPCR analysis validated that the induction of HSPA5 (GRP78), HSPA6, and HSPA1A were significantly increased through CARF knockdown (Figure 5C). Western blotting further confirmed the protein abundance of GRP78, IRE1α, and CHOP (Figure 5D,E) in CARF-depleted HepG2 cells. To substantiate this observation, we silenced CARF expression in AML12 cells and found that the expression of GRP78 and CHOP, but not IRE1a, increased in CARF-depleted AML12 cells (Figure 5F,G). We further found that overexpression of CARF blunted the palmitate-induced ER stress response in HepG2 cells (Figure 5H,I). Compared to vector-transfected control cells, the expression of IRE1α, GRP78, and CHOP was much lower in CARF-overexpressed HepG2 cells. To determine whether deficiency of CARF induces chronic ER stress leading to hepatic steatosis, we inhibited ER stress in CARF-depleted cells through treatment with TUDCA, a chemical chaperone capable of neutralizing ER stress [34,35]. Results shown in Figure 5J indicate that inhibition of ER stress through TUDCA reduced fat accumulation in CARF-depleted cells. In addition to the ER stress signaling pathway, genes associated with calcium signaling pathways were also affected (Appendix A), suggesting that ER function as a reservoir of Ca^2+^ was compromised in CARF-depleted cells, which could also exacerbate ER stress. Collectively, these findings suggest that deficiency of CARF activates chronic ER stress response, leading to lipogenesis in HepG2 cells. 

### 3.5. Knockdown of CARF Induces Oxidative Stress and Cell Death

Oxidative stress is generated due to an imbalance between reactive oxygen species (ROS) generation and antioxidant defenses, which can be caused either by exacerbation of pro-oxidant products or impaired antioxidant machineries [36]. Our RNA seq data further revealed that cellular antioxidant genes Gpx3 and TXNRD3 were significantly reduced in CARF-deficient HepG2 cells (Table 1). Western-blot analyses confirmed that expression of GPX3 decreased in CARF-depleted cells (Figure 6A,B). This result validated the RNA seq data, pointing to the critical role of CARF in the regulation of the cellular antioxidant system. We also measured ROS in HepG2 cells 72 h after CARF knockdown by incubating cells with non-fluorescent H2DCFDA that was oxidized to green-fluorescent DCF via cellular ROS. ROS levels significantly increased in the absence of CARF in HepG2 cells (Figure 6C,D). Furthermore, we found that the expression of Hemoxygenes-1 (HO1) was induced in CARF-depleted cells (Table 1 and Figure 6A,B), which can be interpreted as the cellular response to oxidative stress [37]. Additionally, we performed TUNEL assay to detect apoptosis and found that compared to Sh-SCR, cell death significantly increased in the absence of CARF (Figure 6E,F). Furthermore, to show that CARF-overexpression can protect palmitate-induced oxidative stress and cell death, control- and CARF-overexpressed HepG2 cells were treated with different concentration of palmitate for 24 h, and cleaved PARP1 was tested using Western blot. We found that the amount of cleaved PARP1 was much lower in CARF-overexpressed cells compared to vector control cells (Figure 6G,H), indicating that CARF plays a protective role against oxidative stress and associated apoptosis. Taken together, these results suggest that inhibition of CARF in HepG2 cells enhanced oxidative stress and induced cell death. 

### 3.6. Exogenous Expression of CARF Attenuates HFD-Induced Hepatic Steatosis in Mice

Reduction in CARF in both in vivo and in vitro cellular models of NAFLD indicated that inhibition of CARF triggers fat accumulation in the liver. Next, we sought to assess whether hepatic CARF overexpression (CARF-OE) suppresses the HFD-induced hepatic steatosis in mice. We used AAV-9-based expression vector to achieve hepatic overexpression of CARF [19]. After 16 weeks, HFD mice were sacrificed, and livers were collected for further analysis. No significant changes in bodyweight were found between vector-control and CARF-OE mice (Figure 7A). We performed Western blotting analysis of liver samples to show that an approximately 2-fold overexpression of CARF in livers was achieved (Figure 7B,C). Histological analysis using H&E staining of formalin-fixed and paraffin-embedded liver sections showed that lipid deposition was profoundly reduced in CARF-OE mice compared to livers from vector-control mice (Figure 7D). Consistent with this observation, analysis of prevalence of microvesicular and macrovesicular steatosis demonstrated that, compared to AAV-control livers, the NAFLD score significantly decreased in AAV-CARF livers (Figure 7E). In agreement with this, compared to vector control liver samples, the triglyceride level in CARF-OE livers significantly reduced (Figure 7F). Taken together, we conclude that CARF has the potential to protect against fat deposition in the liver. 

### 3.7. CARF-OE Improves ER Function, Decreases Oxidative Stress, and Improves Insulin Sensitivity in HFD-Fed Mice

Since dysregulation of ER functions plays a crucial role in the onset of hepatic steatosis [38,39], we sought to show whether CARF overexpression in HFD-fed mice can reverse the expression of the ER-stress-responsive gene. In comparison to the control, the expression of GRP78 and CHOP was significantly reduced in CARF-OE livers (Figure 8A–C), indicating that ER stress was attenuated in CARF-OE livers. We also found that expression of GPX3 and HO1 increased and decreased, respectively, in the livers of CARF-OE mice on an HFD (Figure 8D–F), suggesting that CARF can potentially suppress HFD-induced oxidative stress. Furthermore, because hepatic steatosis is associated with IR, we evaluated the effect of CARF-OE on glucose tolerance through serum insulin tolerance and glucose tolerance tests after 14 weeks of feeding on an HFD. Our results showed that, in comparison to control mice, CARF overexpression significantly improved insulin and glucose tolerance in mice on an HFD (Figure 8G–J). Overall, these data suggest that by regulating HFD-induced induced hepatic ER stress and oxidative stress, CARF plays a pivotal role in protecting hepatic steatosis (Figure 8K). 

## 4. Discussion

In this study, we have shown that CARF is a negative regulator of fat accumulation in mouse livers in vivo and in hepatocytes in vitro. Using DIO mice, we demonstrated that, in response to overloaded lipid stress, the level of hepatic CARF decreased along with the downregulation of Sirt1, pAMPK, and pACC, which are causal factors for hepatic steatosis [29,40]. Similarly, using in vitro cellular models of steatosis, we provide evidence that palmitate-induced lipotoxic stress leads to downregulation of CARF, suggesting that excessive FFA accumulation triggers the inhibition of CARF in hepatocytes. These findings led us to hypothesize that CARF protects hepatocytes from fat accumulation. Using the loss of function approach, we found that TG level increased after the silencing of CARF in HepG2 and AML12 cells. In contrast, overexpression of CARF reduced ectopic fat accumulation in vivo mouse livers and in vitro HepG2 cells. 

To understand the underlying mechanisms of how CARF protects against steatosis, we compared the global gene expression profile of CARF-depleted HepG2 cells with the scramble-treated control cells. Functional analysis and GSEA of DEGs evidenced that genes associated with the ER stress or UPR signaling pathway were profoundly affected by silencing of CARF in HepG2 cells. The UPR activation in hepatocytes are orchestrated by three stressor families of proteins, PERK, IRE1α, and ATF6 [41]. Activation of IRE1α increases the splicing of X-box-binding protein-1 (XBP1s) mRNA and subsequent expression of molecular chaperones, e.g., GRP78 and genes associated with ERAD, such as ER degradation-enhancing α-like protein [EDEM] [42]. In this study, we demonstrated that the reduction of CARF in hepatocytes promotes the expression of many of the genes involved in UPR, suggesting that CARF is a vital regulator of ER stress. Since ER stress contributes to de novo lipogenesis or impairs the VLDL-mediated secretion of lipid from the hepatocytes [42], we posit that deficiency of CARF induces fat accumulation mechanistically via activation of UPR. Intriguingly, our study further revealed that the expression of CHOP and rate of apoptosis increased in CARF-deficient cells, suggesting that prolonged ER stress in the absence of CARF leads to apoptosis and hepatocytes injury. In this study, using our established in vivo as well as in vitro models of hepatic steatosis, we elucidated some of the molecular mechanisms by which CARF mitigates HFD-induced hepatic fat accumulation.

In this study, we also showed that knockdown of CARF increased ROS production in hepatocytes. Enhanced apoptosis could also corroborate the increased oxidative stress in CARF-depleted hepatocytes. Cellular ROS production depends on the mitochondrial functions and the balance between pro-oxidant and antioxidant systems, which can be altered via mitochondrial dysfunctions leading to ROS accumulation (55). We observed that GPX3 (glutathione reductase 3) and TXNDR3 (thioredoxin reductase 3), two significant antioxidant genes, were reduced in CARF-deficient hepatic cells. This finding suggests that the reduction of CARF impairs the cellular antioxidant mechanisms that lead to ROS accumulation in hepatocytes. In support of this notion, we found that inhibition of CARF via Sh-RNA accentuated oxidative stress in HepG2 cells. Furthermore, our study showed that HMOX1 (heme oxygenase 1) or HO1, one of the oxidative stress biomarkers, increased following the depletion of CARF. This finding suggests that the upregulation of HO1 might be an adaptive response to protect cells from oxidative stress induced by the inhibition of CARF [43,44]. Furthermore, our study demonstrated that the downregulation of CARF triggers a higher incidence of apoptosis, which could be associated with exacerbated oxidative stress in CARF-depleted cells. In support of these in vitro observations, our in vivo study showed that GPX3 increased, but HO1 expression decreased in the CARF-overexpressing mice fed on an HFD. Furthermore, decreased hepatic apoptosis in CARF-overexpressing livers indicates that the loss of function of CARF in response to lipotoxicity triggers ER stress and oxidative stress, leading to a higher incidence of cell death. An increase in hepatocytes apoptosis is commonly found in both human patients of NAFLD and animal models of steatohepatitis [45]. Therefore, our results suggest that downregulation of CARF via FFA-mediated stress provokes ER stress, oxidative stress, and apoptosis that leads to hepatic steatosis. We can conclude that CARF is a critical regulator of the development and progression of hepatic steatosis, as depicted in the model (Figure 8L).

## 5. Conclusions 

Altogether, the results of this study suggest that overloaded free fatty acid-induced metabolic stress causes the suppression of CARF in hepatocytes. Deficiency of CARF exerts multiple downstream effects leading to perturbation of ER homeostasis and oxidative stress, resulting in fat deposition in the hepatocytes and development of hepatic steatosis. Gain of function of CARF could alleviate FFA-induced metabolic stress in hepatocytes and associated liver diseases.

## Figures and Tables

**Figure 1 cells-12-01069-f001:**
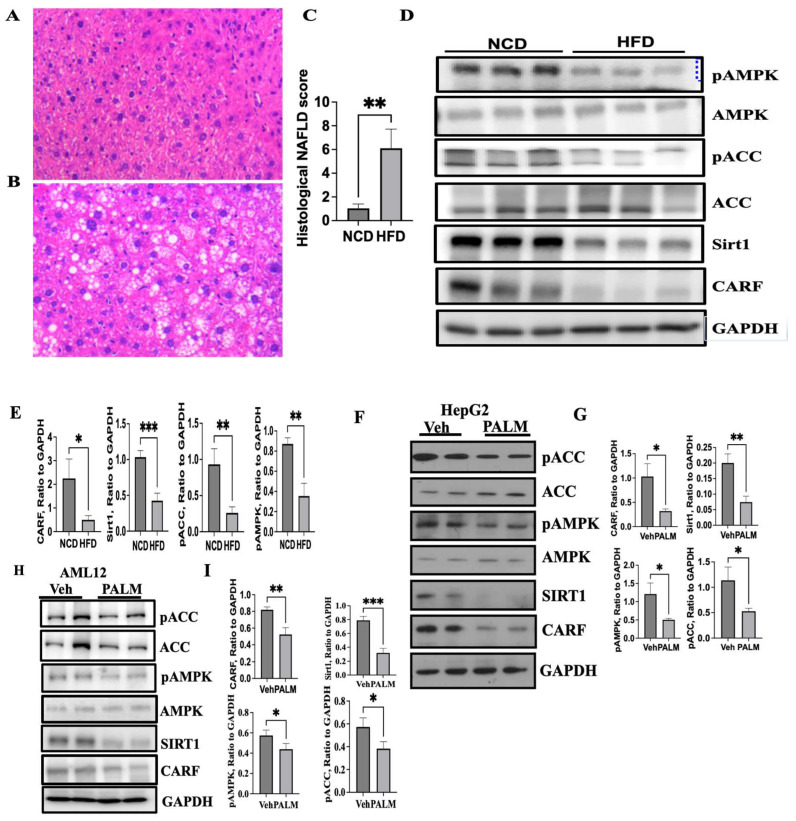
HFD-mediated stress reduces CARF expression in hepatocytes. Representative H&E-stained liver sections from mice (*n* = 6) on an HFD showing a higher level of lipid accumulation (**B**) compared to mice on an NCD (**A**). Quantification of NAFLD intensity as revealed by H&E staining of the liver sections. Values are given as mean ± S.E.M., *n* = 4–5 per group (**C**). Western blotting of liver lysates and its quantification analysis from NCD- and HFD-fed mice (*n* = 5–6 per group) shows a decrease in hepatic CARF expression in mice on an HFD compared to NCD-fed mice (**D**,**E**). Expression of pACC, pAMPK, and Sirt1 are shown as the markers of hepatic steatosis (**D**,**E**). Western blotting and quantification of band intensities (*n* = 3) show the expression of CARF and other markers of steatosis in HepG2 (**F**,**G**) and AML12 (**H**,**I**) cells after exposure to 750 µM palmitate for 24 h (a parametric, unpaired Student’s *t*-test was used to compare between the two groups, * *p* < 0.05, ** *p* < 0.01, *** *p* < 0.001; H&E-stained images were taken at 20× objective).

**Figure 2 cells-12-01069-f002:**
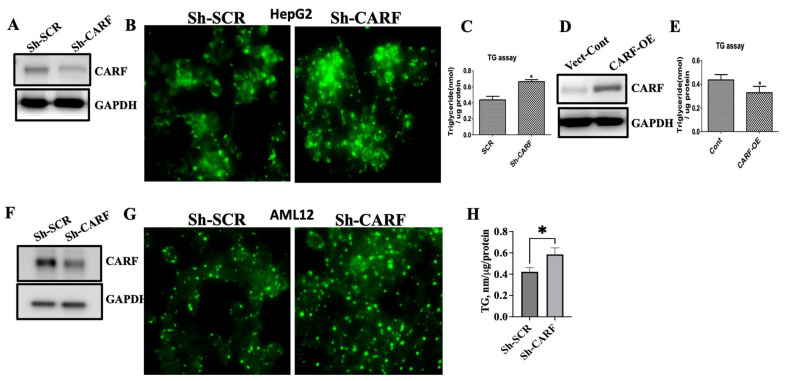
Modulation of CARF regulates lipid accumulation in HepG2 and AML12 cells. Western blot analysis shows CARF expression after Sh-RNA treatment in HepG2 cells (**A**) and AML12 cells (**F**). Fluorescent images of BODIPY-stained HepG2 cells (**B**) and AML12 cells (**G**). TG level after CARF silencing in triplicate cultures of HepG2 cells (**C**) and AML12 cells (**H**). Western blotting showing the overexpression of CARF in HepG2 cells (**D**) and the effect of CARF overexpression on TG accumulation in HepG2 cells from triplicate cultures (**E**). Parametric and unpaired Student’s *t*-test was used for comparison between the two groups. (* *p* < 0.05).

**Figure 3 cells-12-01069-f003:**
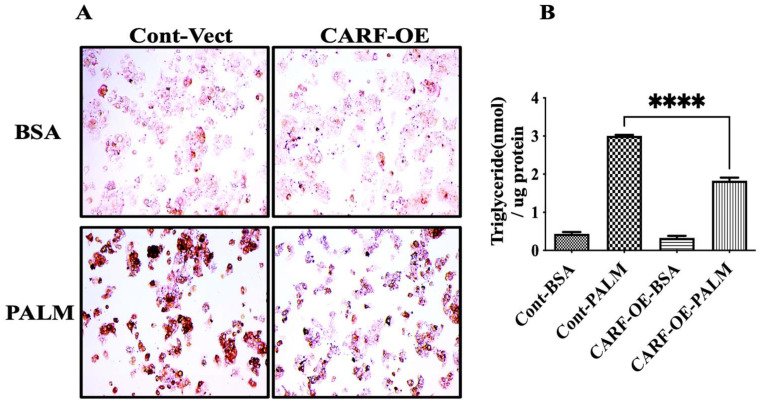
Effect of CARF on ectopic fat accumulation in HepG2 cells. Oil-Red-O staining of HepG2 cells show fat accumulation (**A**). Upper panels show the images of vehicle (BSA)-treated cells, and lower panels show the images of palmitate-treated cells. The left and right panels show the images of vector-control and CARF-overexpressing (CARF-OE) cells, respectively. Oil-Red- O stain shows the reduction of ectopic fat accumulation in CARF-OE cells. (**B**) Shows the level of TG accumulation in HepG2 cells after treating the cells as described in (**A**) (data are shown from 3 independent treatments). One-way ANOVA followed by Holm–Sidak’s multiple comparison test were conducted to show the differences (**** *p* < 0.0001).

**Figure 4 cells-12-01069-f004:**
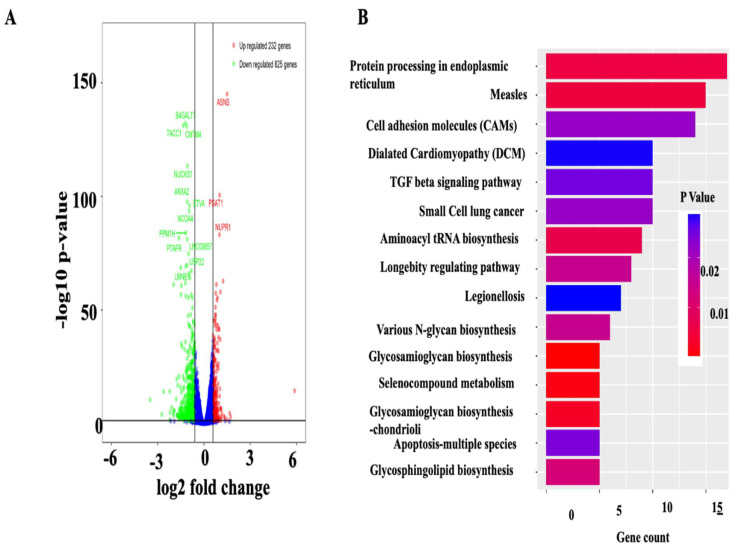
Gene expression profile of HepG2 cells after silencing CARF. Global changes in mRNA expression between SCR- and Sh-CARF-treated HepG2 cells are shown in a volcano plot (**A**). Green dots represent the downregulated differentially expressed genes (DEG), and red dots represent the upregulated DEG in CARF-depleted cells. KEGG analysis of DEGs showing the enrichment of the specific metabolic or signal duction pathways (**B**). The X-axis represents the number of DEGs. The Y-axis represents the KEGG pathways.

**Figure 5 cells-12-01069-f005:**
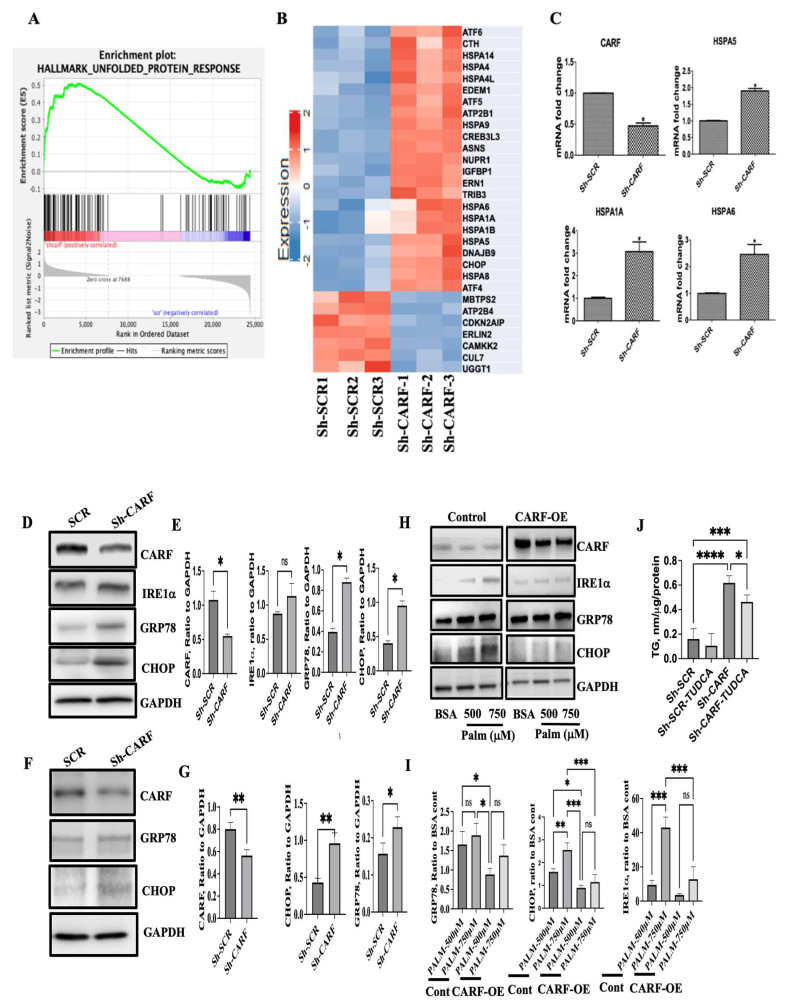
Effect of knockdown of CARF on ER-stress-signaling pathways in HepG2 and AML12 cells. Gene sort enrichment analysis (GSEA) reveals the enrichment of ER stress or UPR-signaling pathways as a regulatory target of CARF in HepG2 cells (**A**). Hierarchical cluster analysis of DEGs enriched in ER-stress- or UPR-signaling pathways (fold changes > 1.5) between SCR- and Sh-CARF-treated cells (**B**). Validation of the GSEA at the mRNA levels of indicated genes revealed through RT-PCR (**C**). Data are shown from 3 independent experiments as mean +/−SD (* *p* < 0.05). Western blotting and its quantification analysis (*n* = 3) showing the protein levels of indicated genes in HepG2 cells 72 h after knockdown of CARF (**D**,**E**). Western blotting and quantification analysis (*n* = 3) of ER-stress marker proteins after CARF knockdown in AML12 cells (**F**,**G**). Effect of overexpression of CARF on palmitate-induced ER stress in HepG2 cells. Western blotting was performed for indicated genes as an ER stressor (**H**). Its quantification from 3 independent experiments (I). GAPDH was used as loading control in all cases. Attenuation of TG accumulation through TUDCA (1 mM) in CARF-deficient HepG2 cells measured using TG assay (**J**). Parametric and unpaired Student’s *t*-test was used to compare between two groups, and one-way ANOVA with Holm–Sidak’s multiple comparison method were used for multiple groups. (* *p* < 0.05, ** *p* < 0.01. *** *p* < 0.001, **** *p* < 0.0001, ns = non-significant).

**Figure 6 cells-12-01069-f006:**
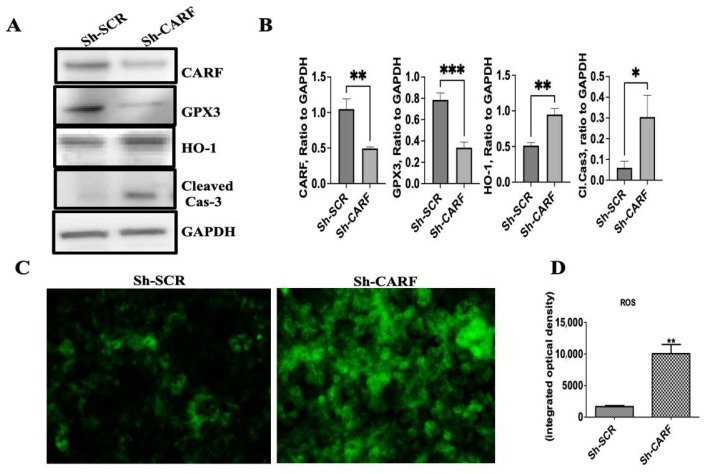
Silencing of CARF induces oxidative stress and apoptosis in HepG2 cells. Western blot analysis of indicated genes shows the effect of CARF knockdown on oxidative stress in HepG2 cells. GAPDH was used as the loading control (**A**). Quantification of Western blotting bands from 3 independent experiments (* *p* < 0.05, ** *p* < 0.01. *** *p* < 0.001) (**B**). HepG2 cells were stained with non-fluorescent H2DCFDA 72 h after knockdown of CARF to show the level of ROS (**C**). The average integrated optical density of the H2DCFDA-stained images (*n* ≥ 5) using Image-Pro software (** *p* < 0.01), (**D**). TUNEL staining after CARF knockdown in HepG2 cells. Upper panels represent Sh-SCR-treated cells and lower panels represent Sh-CARF-treated cells (**E**). Graph showing that ~ 12.8% cells are TUNEL positive for Sh-CARF-treated cells compared to 2.8% in Sh-SCR-treated cells. DAPI positive and TUNEL positive cells from six different microscope fields were counted and percentage of TUNEL positive cells are shown. (*** *p* < 0.001) (**F**). Western blot analysis of PARP1 gene to show CARF overexpression reduces the palmitate-induced apoptosis (**G**) Quantification of cleaved PARP1 from 3 independent experiments. (**H**). Parametric, unpaired Student’s *t*-test and one-way ANOVA with Holm–Sidak’s multiple comparison method were used to compare between two groups and multiple groups, respectively. (** *p* < 0.01).

**Figure 7 cells-12-01069-f007:**
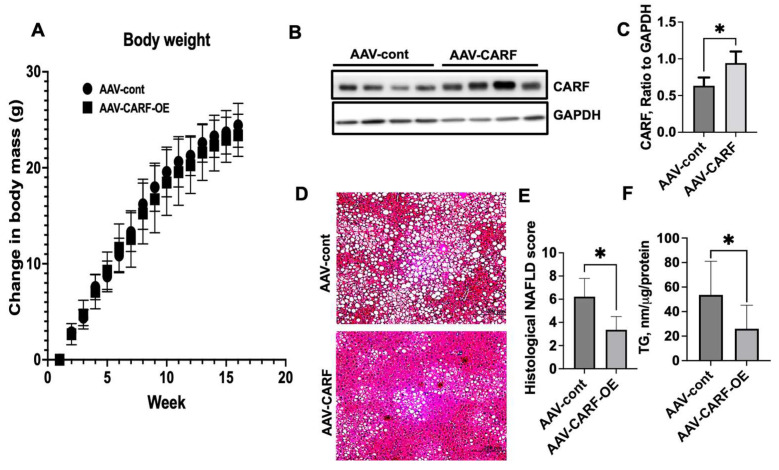
Exogenous expression of CARF attenuates lipid accumulation in liver. Male C57BL6 mice (7 weeks of age) were fed on an HFD for 16 weeks. Control and CARF-overexpressed (CARF-OE) mice (*n* = 8) were transduced with AAV-9 particles expressing empty vector and CARF, respectively, at the 1st week of HFD feeding. Body weight was recorded weekly and compared with the control mice (**A**). Representative Western blotting images of CARF expression in the livers from control and CARF-OE mice (**B**). Quantitation of band intensities of CARF compared with GAPDH (**C**). Representative images of H&E-stained sections of livers collected from control and CARF-OE mice (**D**). Quantification of NAFLD score as revealed through H&E staining of the liver sections. Values are given as mean ± S.E.M., *n* = 4–5 per group (**E**). Total triglyceride level in the livers in the CARF-OE mice compared with control mice (**F**). Parametric, unpaired Student’s *t*-test was applied to compare between two groups. (* *p* < 0.05.) (* H&E-stained images were taken at 20× objective).

**Figure 8 cells-12-01069-f008:**
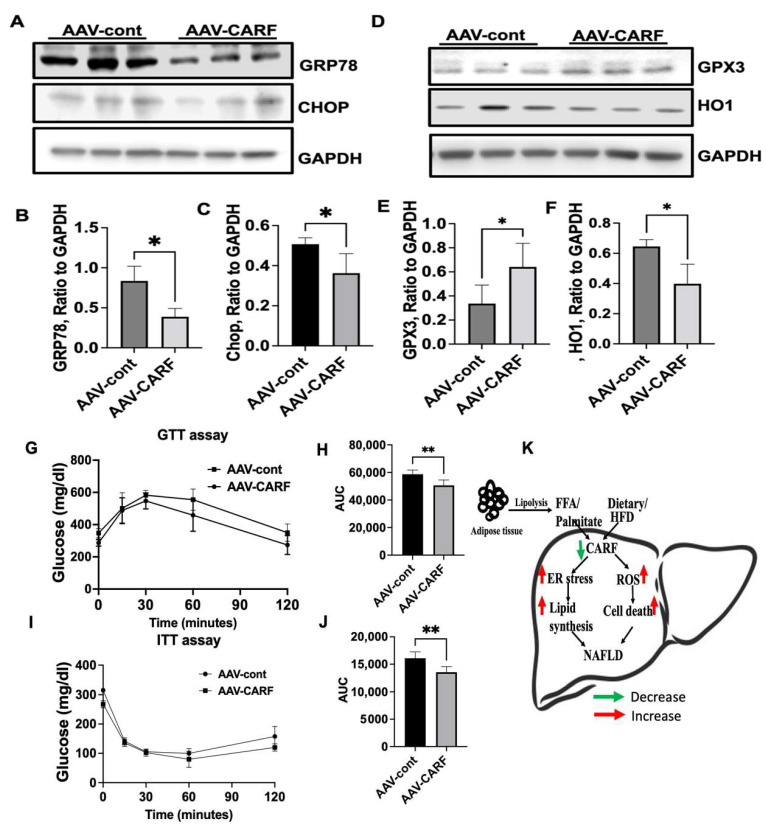
CARF overexpression in HFD-fed mice reduces ER stress, oxidative stress, and improves glucose tolerance. Representative Western blotting analysis of ER-stressor protein in livers from control and CARF-OE mice (**A**). Quantitation of band intensities compared with GAPDH, GRP78 (**B**), and CHOP (**C**). Representatives of Western blotting analysis of oxidative stress and apoptosis marker proteins in the livers from control and CARF-OE mice (**D**). Quantitation of band intensities compared with GAPDH, GPX3 (**E**), and HO1 (**F**). IP-GTT after 14 weeks HFD feeding (**G**,**H**) and IP-ITT after 14 weeks HFD feeding (**I**,**J**). Schematic model showing how the accumulation of dietary lipid or free fatty acids (derived from lipolysis) reduces CARF expression in hepatocytes. Downregulation of CARF induces ER stress and oxidative stress leading to TG accumulation and development of hepatic steatosis (**K**). Parametric, unpaired Student’s *t*-test was used for comparison between two groups. (* *p* < 0.05. ** *p* < 0.01).

**Table 1 cells-12-01069-t001:** ROS signaling associated genes.

Gene Symbol	Name	Fold Change	*p* Value
GPX2	Glutathione peroxidase 2	−1.35	3.52 × 10^−23^
GPX3	Glutathione peroxidase 3	−1.6	4.97 × 10^−33^
HMOX1	Heme Oxygenase 1	2.09	1.03 × 10^−58^
TXNRD3	Thioredoxin reductase 3	−2.35	5.92 × 10^−13^

## Data Availability

All supporting information is available from the corresponding author on reasonable request.

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
