# Peer review of "Fatty Acid Excess Dysregulates CARF to Initiate the Development of Hepatic Steatosis"

_cells, 2023, doi:10.3390/cells12071069_

Round 1
Reviewer 1 Report
The current study unexpectedly found that CARF was suppressed in NAFLD mice and elucidated its underlying mechanism and CARF functions in hepatic steatosis using overexpression or knockdown system in vivo and in vitro system. To understand underlying mechanism of CARFs in liver steatosis, authors did unsupervised transcriptome analysis in knockdown of CARF in HepG2 cells. It showed that ER stress-mediated genes were highly upregulated and it was connected to de novo lipogenesis. It seems quite interesting findings observed in vivo and in vitro system. However, current version of manuscripts describes potential CARF functions in many ways, but it needs discuss more detail especially, how suppression of CARF regulates balancing apoptosis and lipid deposition in same cells so that current version of manuscript needs to be improved.
1) Currently this study heavily relied on in vitro system using only one cell line for HepG2 cells. HepG2 cells is a well-known hepatoma cell line. It should be used other cells to validate CARF functions and it could be better to use primary hepatocytes or normal hepatocytes cell lines to elucidate CARFs functions.
2) In general, HepG2 cells are supposed to grow multilayers so that it will be difficult to capture images of monolayers or single cells after several days. I wonder how it can grow with monolayers and single cells as like in Figure 2A. It looks like morphology of HepG2 cells are changed.
3) As authors showed silencing CARF or knockdown CARF induces oxidative stress, ER stress and lipid accumulation in HepG2 cells. I wonder how it can happen at same times or is there any sequence each event? And even I wonder how fat-induced reduction of CARF induces apoptosis concurrently in same cells. It needs more detailed explanation about balancing between apoptosis and lipid accumulation.
4) Page 2, line 61: Please delete Carlsbad, USA)
5) Figure 2, control (Sh-SCR) for sh-CARF and control (Vect-Cont) for CARF-OE are different for CARF expression level. Have you load same amount of protein concentration to compare?
6) Oil O-Red Staining should be corrected to Oil Red O staining in current manuscript
7) In vivo and in vitro system could be different. Cell type specific manner.
Reviewer 2 Report
The authors evaluated the role of CARF on non-alcoholic fatty liver (NAFL), using in vivo and in vitro models. In general, the idea is innovative, and the models applied are adequate to answer the questions. Nonetheless, the paper has ethical (euthanasia method is missing), and statistical issues (number of biological and technical replicates is missing, pairwise student t test used for >2groups comparisons, the unjustified use of paired t test). Concerning the experiments, a proper histopathological analysis of liver steatosis (score) is missing in the in vivo experiments. The strategy applied for western blot quantification is dubious, as some (few) markers are quantified, and the majority are not. As the majority of blots presented are merely qualitative experiments – also considering that 5/8 figures display blot images – it is difficult to believe in the conclusions. Except for CARF KO or overexpression in vitro (Fig. 2 B and E), all blots must be quantified using an adequate number of replicates.
MAJOR
1. For me, the paper lacks a proper histopathological analysis of steatosis in Figure 1. A score is encouraged (see paper below). This is a short-term model of steatosis, and the photomicrographs depict microvesicular steatosis (although the higher magnification applied can be deceiving).
Liang, W.; Menke, A.L.; Driessen, A.; Koek, G.H.; Lindeman, J.H.; Stoop, R.; Havekes, L.M.; Kleemann., R.; Van Den Hoek, A.M. Establishment of a general NAFLD scoring system for rodent models and comparison to human liver pathology. PLoS One 2014, 9, 1–17, doi:10.1371/journal.pone.0115922.
2. Please perform Oil red staining for representative photomicrographs in the animal studies as well;
3. Please present quantification data for pACC, pAMPK and Sirt1 in Figure 1, using at least than 6 animals per group (in vivo) or 3 subcultures (in vitro)
4. Please provide quantification data for IRE1a, GRP78, and CHOP in Figure 5, using at least 3 subcultures per group;
5. In Figure 5, the loading control (GAPDH) gels are poor, full of artifacts. Please replace them.
6. Please provide quantification data for GPX3, HO1, and casp3 in Figure 6,, using at least or 3 subcultures per group;
7. In Figure 6, blots for HO1 and casp3 are poor or highly edited (pixelated images);
8. Please provide quantification data for PARP1 and cleaved PARP1 in Figure 6,, using at least or 3 subcultures per group;
9. The suggestions made for Fig.1 also applies for Fig. 7, regarding histology.
MINOR
10. Oleate is more steatogenic in HepG2 that palmitate (Ricchi et al., 2009). Please justify using palmitate instead.
Ricchi, M.; Odoardi, M.R.; Carulli, L.; Anzivino, C.; Ballestri, S.; Pinetti, A.; Fantoni, L.I.; Marra, F.; Bertolotti, M.; Banni, S.; et al. Differential effect of oleic and palmitic acid on lipid accumulation and apoptosis in cultured hepatocytes. J. Gastroenterol. Hepatol. 2009, 24, 830–840, doi:10.1111/j.1440-1746.2008.05733.x.
11. The abstract and introduction miss a clear aim;
12. The method of euthanasia is not described;
13. Why not applying FC>2 for RNASeq experiments? Please Justify
14. Also concerning RNASeq, why not using DESeq2 instead of EdgeR?
15. Authors claim that all data were evaluated using student t test. All data were parametric?
16. Also concerning statistics, Fig5. bears more than 2 groups. Therefore pairwise student t test is not adequate, as it could lead to deceiving results. Please reanalyze using adequate tests.
17. Why using paired T test?? Please justify
18. The number of biological and technical replicates is missing in all experiments. Authors only said the number of animals once (line 78) and only for one group (i.e., How many subcultures were used? How many animals per group? How many times each in vitro experiment was performed?).
19. The exact number of replicates and statistical test applied should be clearly presented in figure footnotes;
20. The concentration of each antibody applied in WD experiments is also missing;
21. In Figures 1 and 7, an scale bar is missing in the representative HE photomicrographs (or indicating in the footnotes which objective is).
Round 2
Reviewer 1 Report
Happy to accept it.
Reviewer 2 Report
Authors addressed most of the suggestions, and the manuscript improved in quality.
Only one detail. Authors claimed that thet "don’t have OCT preserved liver samples to do cryosection needed for Oil-Red-O staining". For OR, samples must be frozen (-80ºC), and then embedded in OCT.